# Interplay of Hydropathy and Heterogeneous Diffusion in the Molecular Transport within Lamellar Lipid Mesophases

**DOI:** 10.3390/pharmaceutics15020573

**Published:** 2023-02-08

**Authors:** Antonio M. Bosch, Salvatore Assenza

**Affiliations:** 1Departamento de Física Teórica de la Materia Condensada, Universidad Autónoma de Madrid, 28049 Madrid, Spain; 2Condensed Matter Physics Center (IFIMAC), Universidad Autónoma de Madrid, 28049 Madrid, Spain; 3Instituto Nicolás Cabrera, Universidad Autónoma de Madrid, 28049 Madrid, Spain

**Keywords:** Brownian dynamics, effective diffusion, potential of mean force, partition coefficient, release setups

## Abstract

Lipid mesophases are being intensively studied as potential candidates for drug-delivery purposes. Extensive experimental characterization has unveiled a wide palette of release features depending on the nature of the host lipids and of the guest molecule, as well as on the environmental conditions. However, only a few simulation works have addressed the matter, which hampers a solid rationalization of the richness of outcomes observed in experiments. Particularly, to date, there are no theoretical works addressing the impact of hydropathy on the transport of a molecule within lipid mesophases, despite the significant fraction of hydrophobic molecules among currently-available drugs. Similarly, the high heterogeneity of water mobility in the nanoscopic channels within lipid mesophases has also been neglected. To fill this gap, we introduce here a minimal model to account for these features in a lamellar geometry, and systematically study the role played by hydropathy and water–mobility heterogeneity by Brownian-dynamics simulations. We unveil a fine interplay between the presence of free-energy barriers, the affinity of the drug for the lipids, and the reduced mobility of water in determining the net molecular transport. More in general, our work is an instance of how multiscale simulations can be fruitfully employed to assist experiments in release systems based on lipid mesophases.

## 1. Introduction

Lipid mesophases are self-organized structures where nanoscopic solvent channels emerge from the self-arrangement of lipids in the host solvent. Despite often maintaining the fluidity of the membrane, these aggregates show spatial periodicity following standard crystallographic space groups. In previous studies, the geometry and topology of lipid mesophases have been shown to depend on various parameters, such as pH, temperature, and lipid–solvent (usually water) ratio [1,2,3,4,5,6,7,8]. This has recently made these systems of significant interest for biotechnological applications in material design and drug delivery, as well as fundamental research on ion pumps, membrane protein crystallization, and cryo-enzymatic reactions [9,10,11,12,13,14,15].

In particular, heterogeneity, reproducibility, and high biocompatibility of lipid mesophases have made them a potential tool for drug and nutraceutical delivery [14]. Moreover, the rich structural landscape of lipid mesophases has recently been shown to be naturally explored during digestion of tryglycerides, which likely impacts the delivery of drugs embedded in such hosts [16]. As a consequence, a large amount of research has been devoted to understanding how the chemical and structural features of lipid mesophases influence their transport properties [5,17,18,19,20,21,22,23,24,25,26,27,28,29,30,31,32,33]. Experimental efforts in characterizing the molecular transport through lipid mesophases have unveiled an extensive palette of delivery performance depending on factors such as the geometry and symmetry of the lipid mesophase, the size of the diffusion domain, the hydropathy of the diffusing molecule, and lipid–solvent composition. While opening many possibilities for devising tailored applications, this rich heterogeneity of relevant features makes it lengthy and costly to fully characterize the role played by each of them via experimental assays. Theoretical considerations and computational simulations can provide a complementary insight into these systems, due to their faster and cheaper implementation, but also because they allow to address the effect of the various factors one at a time, which is not always possible in experiments. Although there is a vast literature focused on the modelling of diffusion in confined environments [34,35] and on the exotic water mobility at the interface with hydrophilic objects [36,37,38,39], little work has studied these topics within the context of lipid mesophases [8,28,29,40].

Strikingly, there is a lack of theoretical studies addressing the influence of molecular affinity for the lipids on the transport of the guest particle. This is a particularly important gap in view of the potential applications of lipid mesophases for drug delivery, since about one third of current drugs show low solubility in water [41,42]. The importance of this aspect in drug delivery is being increasingly acknowledged, for instance by investigations focused on solubility aspects in biorelevant media [43]. In this work, we take a step in understanding the impact of hydropathy by studying the diffusion kinetics of a particle spending a finite amount of time in both the water channels and the lipid bilayers. We analyze the role played by the complex interaction free energy between the guest particle and the lipid molecules, as well as the impact of the reduced mobility of water in the vicinity of lipid heads. The latter is expected to strongly affect the quantitative determination of transport properties, as water with lower mobility extends beyond roughly 1 nm starting from the lipid heads [8,40,44], which is comparable to the overall size of the water nanochannels inside lipid mesophases. We focus on the case of a lamellar arrangement with geometrical parameters obtained from reported experimental data. We find that diffusion in the direction perpendicular to the lipid/water interface is strongly regulated by the free-energy barriers obstaculating particle exchange between the lipid and water phases, while parallel diffusion is determined by the hydropathy of the guest molecule, as quantified by the partition coefficient. Heterogeneity of water mobility enters the picture in quantitatively regulating the large-scale diffusion coefficient.

## 2. Materials and Methods

In a lamellar geometry, the system can be conveniently expressed by considering a reference frame with the *z*-axis oriented along the normal to the lipid/water interface. The diffusion of a guest molecule can be assessed by considering a Brownian dynamics in a periodic free-energy landscape U(z) and with a space-dependent diffusion coefficient D(z). Within this framework, the physico-chemical properties of the particle, as well as its interaction with the solvent and the lipids, are concealed within the profiles U(z) and D(z). Ideally, these functions are obtained by running dedicated atomistic simulations of the system; an example is provided by the amino acids study presented later in the manuscript, where U(z) is implemented according to previously-published results of atomistic simulations. Here, we also consider a physically-sound toy model, which accounts, in a minimalistic fashion, for the main features of a typical system, as detailed below. Although simplistic, this framework enables accounting for the features of the system which are the focus of the present study; for example, in the context of strongly-hydrophilic particles in cubic phases, this approach has enabled estimation of the amount of bound water by comparison of theoretical and experimental diffusion coefficients, obtaining values in agreement with direct experimental observations [28]. However, one always has to keep in mind that various microscopical features are not accounted for, including for instance the impact of electrostatics in the diffusion of charged molecules, or the presence of perturbations in the bilayer such as Helfrich undulations [45].

### 2.1. Model

The hydropathy of the guest molecule is described by considering a periodic potential of mean force U(z) such as the one depicted in Figure 1b (we show only one periodicity), which corresponds to the equation
(1)$$U\left(z\right)=\left\{\begin{array}{cc} \Delta U & \mathrm{if}\left|z\right|\leq l-h\\ \Delta U+8\frac{\Delta U_{b}-\Delta U}{h^{2}}{\left(l-h-\left|z\right|\right)}^{2} & \mathrm{if}\;l-h<\left|z\right|\leq l-\frac{3}{4}h\\ \Delta U_{b}-8\frac{\Delta U_{b}-\Delta U}{h^{2}}{\left(l-\frac{1}{2}h-\left|z\right|\right)}^{2} & \mathrm{if}\;l-\frac{3}{4}h<\left|z\right|\leq l-\frac{1}{2}h\\ \Delta U_{b}-8\frac{\Delta U_{b}}{h^{2}}{\left(l-\frac{1}{2}h-\left|z\right|\right)}^{2} & \mathrm{if}\;l-\frac{1}{2}h<\left|z\right|\leq l-\frac{1}{4}h\\ 8\frac{\Delta U_{b}}{h^{2}}{\left(l-\left|z\right|\right)}^{2} & \mathrm{if}\;l-\frac{1}{4}h<\left|z\right|\leq l\\ 0 & \mathrm{if}\;l<\left|z\right|\leq \frac{a}{2} \end{array}\right.\;.$$

In the previous formula, *a* is the lattice parameter, *l* is the total length of a lipid within the bilayer, and *h* is the size of the lipid head (see Figure 1). In order to set reasonable values for these parameters, we considered a=6.65 nm, which corresponds to fully-hydrated $L_{\alpha }$ lamellar mesophases obtained by water/dipalmitoylphosphatidylcholine mixtures at 43 °C [46]. Moreover, we also set l=2.365 nm and h=1 nm based on the electron-density profile computed in Ref. [47] for the same mixture. As for the energy parameters, ΔU is the free-energy difference between the plateaus corresponding to the lipid tails and the water region. Therefore, ΔU>0 for hydrophilic molecules (such as in Figure 1b), while ΔU<0 in the case of hydrophobicity. The parameter $\Delta U_{b}$ introduces a barrier in correspondence with the lipid heads, which can mimic the kinetic barriers associated with the permeability of the membrane; moreover, $\Delta U_{b}$ can be employed to introduce depletion of molecules from the lipid heads ($\Delta U_{b}>0$) or the tendency to sit at the water/lipid interface typical of amphiphilic molecules, which is the case for many proteins ($\Delta U_{b}<0$) [48]. The use of parabolic fragments in Equation (Equation 1) allows for tuning the potential between 0, ΔU and $\Delta U_{b}$, while ensuring that both U(z) and its derivative are continuous throughout space (see Figure 1b), which avoids undesirable numerical instabilities in the simulations.

A similar approach was used to account for heterogeneity in molecular transport. To this aim, we introduced a space-dependent diffusion coefficient D(z) (Figure 1c):(2)$$D\left(z\right)=\left\{\begin{array}{cc} D_{lip} & \mathrm{if}\left|z\right|\leq l\\ D_{lip}+2\frac{D_{wat}-D_{lip}}{w^{2}}{\left(l-\left|z\right|\right)}^{2} & \mathrm{if}\;l<\left|z\right|\leq l+\frac{1}{2}w\\ D_{wat}-2\frac{D_{wat}-D_{lip}}{w^{2}}{\left(l+w-\left|z\right|\right)}^{2} & \mathrm{if}\;l+\frac{1}{2}w<\left|z\right|\leq l+w\\ D_{wat} & \mathrm{if}\;l+w<\left|z\right|\leq \frac{1}{2}a \end{array}\right.\;.$$

In the previous formula, $D_{wat}$ and $D_{lip}$ correspond to the diffusion coefficients of the guest molecule when considered in pure water and in the lipid bilayer, respectively, while *w* is the thickness of the water layer in which the continuous change between $D_{lip}$ and $D_{wat}$ takes place; therefore, *w* accounts for the reduced mobility of water molecules in the vicinity of the lipid heads [44].

Typical values of $D_{wat}$ for nanoscopic objects are found in the range $10^{-10}–10^{-9}$ m^2^/s. For instance, at 25 °C, one has $D_{wat}=0.7-0.9\times 10^{-9}$ m^2^/s for amino acids [49,50,51,52,53], and $D_{wat}=0.5,0.7,0.8\times 10^{-9}$ m^2^/s for ibuprofen [54], aspirin [55], and paracetamol [56], respectively. The value of $D_{wat}$ is expected to be dependent on temperature, *T*. When small temperature differences are considered (such as estimation of $D_{wat}$ at physiological temperature starting from room-temperature measurements), a simple yet effective approach to estimate the effect of *T* is to assume a Stokes–Einstein relation $D_{wat}=k_{B}T/\left(6\pi \eta \left(T\right)R\right)$, where $k_{B}$ is Boltzmann’s constant, *R* is the size of the particle, and η(T) is the temperature-dependent viscosity of water. This approach has enabled accurate predictions of transport of glucose molecules in monolinolein-based cubic phases [57]. Unless stated otherwise, in our simulations, we consider $D_{wat}=0.7\times 10^{-9}$ m^2^/s.

As for the diffusion coefficient in the lipid phase, $D_{lip}$, one expects its value to be significantly smaller than $D_{wat}$ due to the lower fluidity of the lipid membrane as compared to water. For instance, the three-dimensional self-diffusion of lipids for various monoacylglycerols with cubic symmetry has been reported to be $1.1–1.3\times 10^{-11}$ m^2^/s [26], which gives values in the range $1.7–2\times 10^{-11}$ m^2^/s for the lateral diffusion coefficient when accounting for the geometric constraint imposed by the minimal surface at the mid-plane of the lipid bilayer [58]. Amino acids and drugs such as the ones mentioned above are smaller than lipid molecules, so that $D_{lip}$ is expected to be somewhat larger for them. Here, we fix $D_{lip}=0.09D_{wat}$, based on molecular dynamics simulations of paracetamol in DPPC [47].

Finally, the parameter *w* was set in accordance with experimental evidence and molecular dynamics simulations, which point to the existence of 3–4 layers of water with reduced mobility in proximity of the lipid heads [8,40,44]. The specific value of this thickness was selected to be w=0.96 nm (Figure 1c), in order to ensure that $D_{wat}$ is reached exactly at z=a/2, thus avoiding a discontinuity in the derivative of D(z), which would have occurred for larger values of *w*.

### 2.2. Brownian Dynamics

Based on the model introduced above, we run Brownian Dynamics simulations to study the diffusion kinetics. Despite the one-dimensional nature of the potential U(z) and the diffusion coefficient D(z), we integrated the motion in three dimensions, thus also including the movement along planes parallel to the lipid/water interface. This was motivated by the expected effect of a space-dependent diffusion coefficient also on lateral diffusion. Following standard Euler integration, and including a drift term to correctly implement the spatial dependence of the diffusion coefficient [59,60], the updating rule for the position $r\equiv \left(x,y,z\right)$ of a particle is:(3)$$\left\{\begin{array}{c} x\left(t+dt\right)=x\left(t\right)+\sqrt{2Ddt}\;\xi_{x},\\ y\left(t+dt\right)=y\left(t\right)+\sqrt{2Ddt}\;\xi_{y},\\ z\left(t+dt\right)=z\left(t\right)-\frac{D}{k_{B}T}\frac{dU}{dz}dt+\frac{dD}{dz}dt+\sqrt{2Ddt}\;\xi_{z}. \end{array}\right.$$

In the previous formula, *t* is time and dt is the integration timestep, while $\xi_{x},\xi_{y}$ and $\xi_{z}$ are random variables distributed according to a Gaussian function with zero average and unit variance. Although molecular dynamics simulations indicate that diffusion in parallel and perpendicular directions to the lipid/water interface are distinct [8,39], for simplicity in Equation (Equation 3), we consider, for a given position, the same diffusion coefficient for the random movement in any direction. More accurate quantitative estimations will require a proper account of this feature for the system under study.

The choice of length units (σ=1 nm) together with the value chosen for $D_{wat}$ determines a “natural” simulation timescale $\tau =\sigma^{2}/D_{wat}$ for the time *t*. For instance, if $D_{wat}=0.7\times 10^{-9}$ m^2^/s = 0.7 nm^2^/ns, one has τ≃1.4 ns. The value of τ is important in the determination of the integration of the timestep dt. Our rationale in its choice was to consider the largest possible value of dt which correctly recovers the equilibrium distribution of a collection of particles. In this regard, for the system corresponding to U(z) and D(z) as in Figure 1, we performed simulations of $10^{4}$ particles initially located at z=0 for a total time $2\times 10^{4}\tau$, by considering several values of dt; the optimal choice turned out to be $dt=3\times 10^{-4}\tau$, which at the end of the simulation led to the correct sampling of the implemented U(z) (orange points Figure 1b).

For each study reported in Section 3, we performed simulations for ensembles of $10^{3}$ particles up to times ranging between $5\times 10^{3}\tau$ and $5\times 10^{5}\tau$. Particles were pre-equilibrated by randomly extracting their initial position from their Boltzmann distribution by using the Ziggurat algorithm [61]. The value of the total simulation time was adapted according to the system under study, to ensure that the mean-square displacement in the perpendicular direction reached at least 100 nm^2^. We found that this threshold value was sufficient to ensure a good estimation of the long-time diffusion coefficient.

The diffusion kinetics of the system was monitored by computing the mean-square displacement:(4)$$\left\{\begin{array}{c} MSD_{\|}\left(t\right)=\langle {\left(x\left(t\right)-x_{0}\right)}^{2}\rangle +\langle {\left(y\left(t\right)-y_{0}\right)}^{2}\rangle \;\\ MSD_{⊥}\left(t\right)=\langle {\left(z\left(t\right)-z_{0}\right)}^{2}\rangle \;. \end{array}\right.$$

In the previous formula, $r_{0}\equiv \left\{x_{0},y_{0},z_{0}\right\}$ is the initial position of each particle, while $\langle \cdots \rangle$ denotes averaging over the whole ensemble. $MSD_{\|}\left(t\right)$ and $MSD_{⊥}\left(t\right)$ describe the diffusion kinetics in the directions parallel and perpendicular to the lipid/water interface, respectively. At long times, in both cases, the MSD is expected to be purely diffusive [28]: $MSD_{\|}\left(t\right)=4D_{\|}t$ and $MSD_{⊥}\left(t\right)=2D_{⊥}t$, where $D_{\|}$ and $D_{⊥}$ are the effective diffusion coefficients. The different prefactors account for the different dimensionalities of the two diffusion kinetics (two and one dimensions, respectively).

### 2.3. Relationship between logP and $\langle D\rangle$

The logarithm of the partition coefficient logP is defined as $\log P=\log_{10}\left(c_{lip}/c_{wat}\right)$, where $c_{lip}$ and $c_{wat}$ are the concentrations of particles in the lipid and water phases [62]. Denoting as *l* the length of the lipids and as *a* the lattice parameter, one has in general
(5)$$10^{\log P}=\frac{c_{lip}}{c_{wat}}=\frac{\int_{0}^{l}e^{-\frac{U(z)}{k_{B}T}}dz/l}{\int_{l}^{\frac{a}{2}}e^{-\frac{U(z)}{k_{B}T}}dz/\left(\frac{a}{2}-l\right)}\;$$
where we consider only half repeating units due to the symmetry of the system. Note that this formula is not restricted to our choice of U(z) operated above; instead, it holds, in general, under the approximation that one can sharply distinguish the lipid and water phases. As a further approximation, we assume that within the water phase U(z)=0. This neglects desolvation effects on the guest molecule within the water layer in contact with the lipid heads, as well as electrostatic interactions. Nevertheless, actual computation of U(z) from atomistic simulations suggests that this assumption is pretty reasonable (see, e.g., Figure 7a). Within this approximation, the denominator in the previous formula is 1, thus yielding
(6)$$\int_{0}^{l}e^{-\frac{U(z)}{k_{B}T}}dz=l\;10^{\log P}\;.$$

The average diffusion coefficient $\langle D\rangle$ is obtained by thermal averaging of D(z):(7)$$\langle D\rangle =\frac{\int_{0}^{\frac{a}{2}}D\left(z\right)e^{-\frac{U(z)}{k_{B}T}}dz}{\int_{0}^{\frac{a}{2}}e^{-\frac{U(z)}{k_{B}T}}dz}\;.$$

The denominator in the previous formula can be rearranged by means of Equation (Equation 6) as
(8)$$\int_{0}^{\frac{a}{2}}e^{-\frac{U(z)}{k_{B}T}}dz=\int_{0}^{l}e^{-\frac{U(z)}{k_{B}T}}dz+\int_{l}^{\frac{a}{2}}e^{-\frac{U(z)}{k_{B}T}}dz=l\;10^{\log P}+\left(\frac{a}{2}-l\right)\;.$$

Similarly, the numerator in Equation (Equation 7) can be rearranged as
(9)$$\int_{0}^{\frac{a}{2}}D\left(z\right)e^{-\frac{U(z)}{k_{B}T}}dz=\int_{0}^{l}D\left(z\right)e^{-\frac{U(z)}{k_{B}T}}dz+\int_{l}^{l+w}D\left(z\right)e^{-\frac{U(z)}{k_{B}T}}dz+\int_{l+w}^{\frac{a}{2}}D\left(z\right)e^{-\frac{U(z)}{k_{B}T}}dz\;.$$

In the first and third term on the right-hand side of the previous formula, one can assume constant values for the diffusion coefficient equal to $D_{lip}$ and $D_{wat}$, respectively. The second term can be rewritten as $w\;\bar{D}$, where $\bar{D}$ is the average value of D(z) in the layer of water molecules with non-trivial mobility (compare Figure 1c). We assume that $\bar{D}=\left(D_{wat}+D_{lip}\right)/2$, which considers a symmetric profile of D(z) within this region, as is the case for the toy model introduced in Equation (Equation 2). This allows for rewriting the numerator as
(10)$$\int_{0}^{\frac{a}{2}}D\left(z\right)e^{-\frac{U(z)}{k_{B}T}}dz=D_{lip}\int_{0}^{l}e^{-\frac{U(z)}{k_{B}T}}dz+w\frac{D_{wat}+D_{lip}}{2}+\left(\frac{a}{2}-l-w\right)D_{wat}\;$$
that is, by rearranging and making use of Equation (Equation 6),
(11)$$\int_{0}^{\frac{a}{2}}D\left(z\right)e^{-\frac{U(z)}{k_{B}T}}dz=D_{lip}\left(l\;10^{\log P}+\frac{w}{2}\right)+D_{wat}\left(\frac{a}{2}-l-\frac{w}{2}\right)\;.$$

Plugging Equations (Equation 8) and (Equation 11) in Equation (Equation 7) finally yields the formula reported in Equation (Equation 14) in the main text.

### 2.4. Amino Acids Simulations

We simulated the diffusion of 16 amino acids by considering for U(z) the potential of mean force derived in Ref. [63], where these residues were considered in the presence of DOPC. For the four charged amino acids (Arg, Lys, Glu, Asp), we considered $U\left(z\right)=min\{U_{c}\left(z\right),U_{0}\left(z\right)\}$, where $U_{c}\left(z\right)$ and $U_{0}\left(z\right)$ are the free-energy profiles obtained for the charged and neutral variants of the amino acids, shifted according to the free-energy cost of their neutralization in pure water [63]. We deem this approximation to be reasonable, as seen explicitly in constant-pH simulations performed for similar systems [64]. The U(z) profiles are reported in Figure 7a. As for the diffusion profile, we considered Equation (Equation 2) with parameters adapted to the present system. We kept the value a=6.65 nm for the lattice parameter. Following Ref. [63], we assigned to the lipid length the value l=2.5 nm. Accordingly, we set w=0.825 nm in order to attain $D_{wat}$ exactly at z=a/2. The diffusion coefficients in water, $D_{wat}$, were obtained by considering for each residue the values obtained experimentally [49,50,51,52,53,65,66] and subsequently estimating the corresponding values at 37 °C according to the Stokes–Einstein equation, as discussed above [57]. The obtained values range between 0.9 nm^2^/ns (obtained for Trp) and 1.3 nm^2^/ns (obtained for Cys), with an average equal to 1.04 nm^2^/ns. As in the toy model, the value of $D_{lip}$ was set to $D_{lip}=0.09D_{wat}$ for each residue.

Brownian dynamics simulations were performed with the same parameters as discussed above. Due to the large values of the free-energy barriers, the diffusion coefficients for Arg, Asn, Lys, Asp, Gln, and Ile could not be assessed by brute-force simulations. Instead, for each of these systems, we considered various sets of simulations in which U(z) was renormalized by a factor α<1. The value of $D_{⊥}\left(\alpha \right)$ was computed for each simulation set and its dependence on α was fitted via an exponential decay. The sought value for the original system was then obtained by extrapolating to α=1. For Asn, Lys, Asp, Gln, and Ile, we considered α=0.3,0.4,0.5,0.6. In the case of Arg, the strong barrier imposed lower values α=0.15,0.20,0.25,0.30. As for $D_{\|}$, for each value of α, we checked the quantitative agreement between the value obtained from the simulations and the result found by applying Equation (Equation 13); then, we considered the predicted value computed for α=1.

## 3. Results

Periodic lipidic mesophases exist in a wide variety of arrangements, including lamellar, hexagonal, and cubic symmetries [14]. Moreover, for each symmetry, an extended range of geometrical features can be obtained, e.g., channel swelling by addition of co-surfactants [5]. For the lamellar symmetry, the lipids in the bilayer can be arranged in different ways, such as crystals ($L_{c}$ phase), gels ($L_{\beta }$), or fluid membranes ($L_{\alpha }$) [1,46]. As the main goal of this work is to understand the combined effect of heterogeneous diffusion and hydropathy on molecular transport, we focused on the simple case of a lamellar symmetry with fixed geometrical features (Figure 1), leaving the important topic of the impact of topology and geometry to future work. We fixed the lattice parameter a=6.65 nm, the lipid length l=2.365 nm, and the size of lipid heads h=1 nm by following data reported in the literature [46,47] (see Methods for further details). It is expected that quantitative results depend on the choice of these parameters, but the qualitative impact of hydrophobicity and spatial dependence of diffusion on molecular transport does not change once the values of a,l and *h* have been fixed. We note that, for any system of interest, these parameters can be obtained directly from experiments such as small-angle X-ray or neutron scattering; particularly, *a* can be obtained by analyzing the peak positions of the scattering profiles [17], while *l* and l−h correspond to half the thickness of the bilayer and of its hydrophobic part, respectively, which can be computed by suitably fitting the scattering profiles [67] or by employing geometric arguments based on the known composition of the system [1].

In order to establish a connection between the microscopic insights gathered by molecular dynamics simulations and the macroscopic transport properties relevant for experiments and practical applications, it is necessary to introduce a mesoscopic description of lipid mesophases, which takes as input the microscopic details of the system and provides predictions on the macroscopic diffusion. We can describe the heterogeneous environment offered by lipid mesophases at the nanoscopic scale by introducing a periodic potential of mean force U(z) and a space-dependent diffusion coefficient D(z), both depending on the position *z* of the particle with respect to the the lipid–water interface (Figure 1). These features can be extracted from molecular dynamics simulations or, although only partially, from experimental data. To assess the impact of hydropathy, we first consider a minimalistic model of U(z), for which we report a representative plot for a single periodicity in Figure 1b. The parameter ΔU gives the overall change in free energy upon inclusion of the guest molecule in the lipid membrane as compared to water, while $\Delta U_{b}$ represents the free-energy barrier (or well, for negative values) which regulates the timescale of particle exchange between the lipid and water regions. Heterogeneous transport is captured by a position-dependent diffusion coefficient D(z) (Figure 1c), which continuously changes from $D_{lip}$ in the lipid membrane to $D_{wat}$ in water, with the change happening within a distance range of about w=1 nm, as reported in the literature [8,40,44] (see Methods for further discussion on the matter). U(z) and D(z) were then employed to run Brownian Dynamics simulations aimed at assessing the diffusion kinetics of the system at large scales. Full details of the model and of the simulation setup are described in the Methods. In a later section, we also consider a practical case study focused on amino acids, for which more realistic potentials were extracted from molecular dynamics simulations reported in the literature.

### 3.1. Assessing the Importance of Each Physical Ingredient

#### 3.1.1. Impact of $\Delta U_{b}$

The simulation setup enables devising systems which, albeit unrealistic, provide clear insights on the impact of each feature taken separately. In this section, we focus on how the barrier $\Delta U_{b}$ affects large-scale diffusion. To this aim, we thus fix ΔU=0 and $D_{lip}=D_{wat}=0.7$ nm^2^/ns, so as to isolate the effect of $\Delta U_{b}$ alone. Examples of potentials U(z) are reported in the insets of Figure 2. In the figure, the main plots show the mean-square displacement (MSD) for selected values of $\Delta U_{b}$, in order to highlight the role played by the magnitude and sign of this parameter. Particularly, in Figure 2a, we show the MSD for $\Delta U_{b}=-4k_{B}T$ (green squares) and $\Delta U_{b}=4k_{B}T$ (orange circles). The empty symbols correspond to the MSD computed along planes parallel to the lipid/water interface, MSD_‖_. Since in the considered systems the diffusion coefficient is constant throughout space, lateral diffusion is unaffected by the value of U(z). Therefore, MSD_‖_ shows a standard diffusive behavior characterized by a diffusion coefficient $D_{\|}=D_{wat}$, i.e., MSD_‖_
$=\;4D_{wat}t$ (grey dashed line in Figure 2a). In a log-log plot such as the ones considered in the figure, this corresponds to a linear function with slope one shifted according to the value of $D_{\|}$.

In contrast, the kinetics along *z* (filled symbols in Figure 2a) are characterized by richer dynamics, where three regimes can be identified. At short times (e.g., t≲0.1 ns for the orange circles in Figure 2a), the particles diffuse with a diffusion coefficient $D_{⊥}=D_{wat}$. In contrast, a subdiffusive behavior is detected at intermediate times (0.1 ns ≲t≲10 ns), as evidenced by a local slope lower than one. At longer times (t≳10 ns), standard diffusion is retrieved, but with an effective diffusion coefficient $D_{⊥}<D_{wat}$. Notably, the two sets of data collapse onto the same curve in this diffusive regime, suggesting that the long-time behavior of the system is independent of the sign of $\Delta U_{b}$, although the onset of this regime is shifted towards larger values of *t* for positive values of the barrier. In Figure 2b, we show that similar considerations apply for $\Delta U_{b}=\pm 8k_{B}T$, but the larger magnitude of the barriers results in lower values of the effective diffusion coefficient $D_{⊥}$ along the direction perpendicular to the lipid/water interface.

In Figure 3a, we report the long-time diffusion coefficients obtained by varying the magnitude of $\left|\Delta U_{b}\right|$ in the range $1–10k_{B}T$, for both positive (purple hexagons) and negative (golden stars) values of the barrier. Coherently with Figure 2, the lateral diffusion coefficients $D_{\|}$ (empty symbols) are independent of the height of the barrier, and correspond to the value of $D_{wat}$ (grey dashed line). On the other hand, the perpendicular diffusion coefficients (filled symbols) are strongly affected by $\Delta U_{b}$, displaying an exponential decay starting from $5k_{B}T$, for which a two-parameters best fit yields $D_{⊥}\left({nm}^{2}/ns\right)=8.0\;e^{-0.93\left|\Delta U_{b}\right|/k_{B}T}$ (red continuous line). This can be rationalized by observing that, for large values of $\left|\Delta U_{b}\right|$, jumping events from one side of the barrier to the other are rare. The rate *k* at which these events take place is approximately captured by the Arrhenius equation, $k≃Ae^{-\left|\Delta U_{b}\right|/k_{B}T}$. At time *t* (assumed to be large enough for the system to be found in the long-time diffusing regime), the cumulate number of expected events is kt. Each jump corresponds to a certain length λ within the same order of magnitude of the lattice parameter *a*. The total distance *L* travelled by the random set of jumps thus satisfies the relation $L^{2}=kt\lambda^{2}=Ae^{-\left|\Delta U_{b}\right|/k_{B}T}\lambda^{2}t$. Since one has also $L^{2}=2D_{⊥}t$, one thus obtains $D_{⊥}=\left(A/2\right)e^{-\left|\Delta U_{b}\right|/k_{B}T}\lambda^{2}\propto e^{-\left|\Delta U_{b}\right|/k_{B}T}$. Note that the Arrhenius formula predicts a prefactor equal to −1 in the exponent, which is in good agreement with the best-fitting value. Finally, the results obtained for $\Delta U_{b}>0$ and $\Delta U_{b}<0$ fall on the same master curve, confirming the trend observed in the long-time regime of $MSD_{⊥}$ in Figure 2.

#### 3.1.2. Impact of ΔU

We performed a similar study focused on the hydropathy parameter ΔU, for which a representative profile of U(z) is reported as an inset in Figure 3b. Qualitatively, the trend is very similar to the previous study. This is expected, since the system is being described by a set of periodically-placed barriers with equal heights, which is similar to the case reported in Figure 3a. The same reasoning as in the previous section applies, although the quantitative details of the exponential fit change. In this case, the fitting formula for the perpendicular diffusion coefficient is $D_{⊥}\left({nm}^{2}/ns\right)=2.6\;e^{-0.97\left|\Delta U\right|/k_{B}T}$ (red continuous line in Figure 3b).

#### 3.1.3. Impact of $D_{lip}/D_{wat}$

Finally, we also considered systems with a position dependent diffusion coefficient D(z) as in Figure 1c, but in which no potential was present, i.e., U(z)=0, which corresponds to $\Delta U_{b}=\Delta U=0$. We report the long-time diffusion coefficients $D_{\|}$ (empty symbols) and $D_{⊥}$ (filled symbols) in Figure 4. In contrast with the previous cases, here the *z* dependence of the local diffusion coefficient affects $D_{\|}$, which is found to increase linearly with the ratio $D_{lip}/D_{wat}$. Considering that in the Brownian motion (Equation (Equation 3)) the drift term does not involve directly the parallel direction, it is expected that the long-term value of $D_{\|}$ is the average value of D(z). From Equation (Equation 2), one thus finds
(12)$$D_{\|}=\frac{1}{a}\int_{-\frac{a}{2}}^{\frac{a}{2}}D\left(z\right)dz=D_{wat}-\left(D_{wat}-D_{lip}\right)\frac{2l+w}{a}\;$$
which is reported in Figure 4 as a red continuous line, and quantitatively reproduces the simulation data.

As for the perpendicular direction, from Figure 4, one finds that $D_{⊥}$ is systematically lower than $D_{\|}$. The simulation results can be rationalized by considering the two extreme cases. When $D_{lip}≃D_{wat}$, one retrieves the diffusion coefficient obtained in the trivial case $D\left(z\right)=D_{wat}$, i.e., $D_{⊥}=D_{wat}$ due to the absence of an external potential. In contrast, for $D_{lip}\ll D_{wat}$, one expects that the particles are effectively confined along the perpendicular direction, i.e., that they do not diffuse transversally to the lipid/water interface. Indeed, in order to increase $MSD_{⊥}$, they need to traverse the full periodic repeat of the system; however, within the lipid region, there is practically no diffusion, so that the particle will need a large amount of time to cross it. In other terms, one expects in this limit that the transition to the long-time regime (see, e.g., Figure 2) moves towards larger and larger times, thus resulting in a vanishing value of $D_{⊥}$. Note that this reasoning does not apply to $D_{\|}$, since there is no need for the system to cross the lipid phase in order to increase $MSD_{\|}$. Quantitatively, we found that a good description of the data is obtained by means of a power law $D_{⊥}=D_{wat}{\left(D_{lip}/D_{wat}\right)}^{0.92}$, which provides the correct limiting behavior and is reported as a blue dashed line in Figure 4.

### 3.2. Putting the Physical Ingredients Together

Having assessed the role played by each feature of the model, we now consider their interplay in more complex scenarios. In this regard, we performed simulations with the full potential U(z) (Figure 1a), by varying simultaneously the values of ΔU and $\Delta U_{b}$. We run two sets of simulations: in Set 1, the diffusion coefficient was considered to be constant ($D_{lip}=D_{wat}$), while, in Set 2, we implemented a more realistic profile for D(z), with $D_{lip}=0.09D_{wat}$ (see Methods for further details). The results of these simulations are reported in Figure 5 and Figure 6.

In Figure 5, we report the results obtained for $D_{⊥}$ in the simulations of Set 1 (Figure 5a) and Set 2 (Figure 5b). In both cases, the largest values of $D_{⊥}$ are obtained for lower values of $\left|\Delta U_{b}\right|$ and $\left|\Delta U\right|$, i.e., close to the center of each figure. This is intuitively understandable, as this region corresponds to lower barriers to be overcome. Similarly to what is observed in Figure 3, increasing the magnitude of $\left|\Delta U_{b}\right|$ and $\left|\Delta U\right|$ has a dramatic effect on $D_{⊥}$, which rapidly decreases to low values (note that, in Figure 5, the scale in the bar is logarithmic). When comparing the results of the two sets of simulations for given values of $\left|\Delta U_{b}\right|$ and $\left|\Delta U\right|$, it is evident that, for Set 2, the value of $D_{⊥}$ is systematically lower than for Set 1. This is also expected, since the average diffusion coefficient $\langle D\rangle$ for the variable case is lower. However, the overall change of $D_{⊥}$ cannot trivially be ascribed to a normalization of the results by $\langle D\rangle$, i.e., in general, $D_{⊥,Set2}\neq D_{⊥,Set1}\cdot \langle D\rangle$.

As for the lateral diffusion coefficient $D_{\|}$, in the simulations of Set 1, one finds trivially that $D_{\|}=D_{wat}$, analogous to what was reported in Figure 3. The results obtained for Set 2 are instead reported in Figure 6a. In this case, the largest values are obtained for $\Delta U_{b}>0,\Delta U>0$ with a large magnitude, i.e., by maximizing the depletion from the lipid phase. This can be understood by observing that, to achieve an efficient lateral diffusion, one does not need the particles to cross a full periodicity, but rather to maximize the time spent in the water phase, characterized by a larger mobility (Figure 1c). This is best achieved by increasing the energy penalty for particle localization in the lipid phase, i.e., by large, positive values of $\Delta U_{b}$ and ΔU (Figure 1b). Quantitatively, as discussed above, the absence of a direct drift term in the parallel direction suggests that $D_{\|}$ can be identified with the average $\langle D\rangle$. Hence, in the presence of a potential U(z), one can generalize Equation (Equation 12) as
(13)$$D_{\|}=\frac{1}{a}\int_{-\frac{a}{2}}^{\frac{a}{2}}e^{-\frac{U(z)}{k_{B}T}}D\left(z\right)dz\;.$$

To enable a direct link with experimentally-measurable quantities, we quantify the relative amount of time spent in the lipid phase by means of the logarithm of the partition coefficient, $\log P=\log_{10}\left(c_{lip}/c_{wat}\right)$ [62], where $c_{lip}$ and $c_{wat}$ are the concentrations in the lipid and water phase. Rewriting Equation (Equation 13) by means of the partition coefficient yields (see Methods),
(14)$$D_{\|}=\frac{D_{lip}\left(l\;10^{\log P}+\frac{w}{2}\right)+D_{wat}\left(\frac{a}{2}-l-\frac{w}{2}\right)}{l\;10^{\log P}+\frac{a}{2}-l}\;.$$

Note that, for strongly hydrophobic molecules, logP is positive and has a large magnitude, so that, in the previous formula, the terms containing the factor $10^{\log P}$ are much larger than the rest. In this case, one thus obtains $D_{\|}≃D_{lip}$, which is expected since the molecule spends virtually all the time within the lipid bilayer. In contrast, for strongly hydrophilic molecules logP<0 and large in magnitude, so that $10^{\log P}≃0$. The value obtained for $D_{\|}$ in this case is the average diffusion coefficient within the water phase, which does not trivially correspond to $D_{wat}$ due to the position dependence of D(z) (Figure 1c). In Figure 6b, we compare the numerical results from the simulations (golden stars) with Equation (Equation 14) (red continuous line). The excellent agreement confirms the quantitative correspondence between $D_{\|}$ and $\langle D\rangle$. To fully appreciate the dependence of $D_{\|}$ on logP, in Figure 6b, we also considered an extended set of systems (purple triangles), with ΔU and $\Delta U_{b}$ going beyond the maximum magnitude $5k_{B}T$ considered in the simulations. For these systems, $D_{\|}$ was computed according to Equation (Equation 13).

### 3.3. Application: Large-Scale Transport of Amino Acids

As a practical example of the usage of the present approach, we dedicate this section to the study of diffusion of amino acids through lamellar phases. The potential of mean force U(z) for amino acids within phospholipidic bilayers has been the focus of previous investigation work [63,68]. Rather than using the toy model proposed in Figure 1b, we thus consider here U(z) for 16 amino acids as computed in Ref. [63], where the authors performed enhanced-sampling molecular dynamics of various residues embedded in DOPC bilayers. The resulting potentials are reported in Figure 7a. Although being more complex, they resemble the toy model introduced in Figure 1b, showing a plateau in correspondence of the lipid tails (*z* close to zero) and barriers or wells in proximity of the lipid heads (|z| around 2–2.5 nm). Albeit being accessible to atomistic simulations [69], the position dependence of the diffusion coefficient was unfortunately not addressed in Ref. [63], so that we consider the toy model for D(z) presented in Figure 1c. Further details of the simulations are provided in the Methods.

Based on our analysis in the previous sections, we expect that the presence of barriers in U(z) has a dramatic effect on the perpendicular diffusion coefficient $D_{⊥}$. To quantify the extent of such barriers, for each amino acid, we computed the minimum and maximum values of the potential of mean force, which we denote as $U_{min}$ and $U_{max}$, respectively. For instance, for arginine, we obtain $U_{min}≃-8.5k_{B}T$ and $U_{max}≃23.5k_{B}T$ (second panel from the top-left corner in Figure 7a). In Figure 7b, we report $D_{⊥}$ as a function of the difference $U_{max}-U_{min}$, which we took as an indicator of the strength of the barriers. The heterogeneity of profiles for U(z) results in extremely different values for $D_{⊥}$, which span as many as ten orders of magnitude (compare main plot and inset in Figure 7b). Moreover, an Arrhenius-like exponential dependence captures the data over all the different scales, with the best-fitting formula being $D_{⊥}≃0.5\cdot e^{-0.77\left(U_{max}-U_{min}\right)/k_{B}T}$ (grey dashed line). The data cluster according to the physico-chemical properties of the amino acids. Particularly, the ones with charged or polar side chains are characterized by the lowest values of $D_{⊥}$ (golden stars and red circles in Figure 7b, respectively). This is not expected a priori, since free-energy barriers are expected also for apolar residues. However, as stressed by the different values of $U_{max}-U_{min}$ obtained for the various sets of amino acids (see also full profiles of U(z) in Figure 7a), the depletion of charged and polar residues from the lipid bilayer is significantly stronger than the free energy gained by embedding apolar amino acids. A closer look to the free-energy profiles indicates that, for apolar residues, localization within the whole lipid region is either energetically favourable or it comes at a negligible cost. The barrier for perpendicular diffusion is thus provided by the depth of the free-energy wells. In contrast, for charged and polar residues, localization in the region corresponding to the lipid tails is highly costly, while the lipid heads are favoured or have low associated cost. Overall, this provides a significantly steeper barrier to perpendicular diffusion.

As for $D_{\|}$, analogous to Figure 6b, in Figure 7c, we plot the results obtained from the simulations as a function of logP. The grey dashed line is the prediction according to Equation (Equation 14), which is again found to quantitatively capture the simulation data. Importantly, $D_{\|}$ is always found within the same order of magnitude ($10^{-1}$ nm^2^/ns), which indicates that $D_{\|}\gg D_{⊥}$ for large enough barriers, hence suggesting parallel diffusion to be dominant in such scenario. Similar to $D_{⊥}$, also in this case, we find a clustering of the points according to the physico-chemical properties of the residues. Particularly, apolar residues (green empty squares) have a stronger affinity for the lipid phase, as denoted by the larger values of logP (taken with sign). This implies that a larger fraction of time is spent in the slowly-diffusing region corresponding to the lipids, thus yielding low values of $D_{\|}$. Similarly, polar residues (red empty circles) have lower affinity for lipids, thus resulting in faster parallel diffusion. Unintuitive results are obtained instead for charged residues (golden empty stars), for which one would imagine a strong depletion from the lipid phase. While Glu (E) and Asp (D) abide by the expected behavior (low affinity for lipids, a large value of $D_{\|}$), quite surprisingly, Lys (K) and Arg (R) show instead the opposite behavior. An inspection of the corresponding U(z) profiles for these residues (Figure 7a) reveals the presence of a deep well ($≃-10k_{B}T$) in correspondence with the lipid heads. Thus, despite being strongly depleted from the lipid tails, Lys and Arg spend most of the time sitting at the lipid/water interface, which is characterized by slow diffusion. In terms of the toy model, one can locate these residues in Figure 6a in correspondence with ΔU>0 and $\Delta U_{b}<0$, both with large magnitudes. However, it is important to stress that the quantitative value of $D_{\|}$ for Lys and Arg (and more in general for all the amino acids with positive values of logP) strongly depends on the profile chosen for D(z) (Figure 1c) and on the value of $D_{lip}$, so that a more precise quantitative estimation requires a proper determination of D(z) from ad hoc atomistic simulations. The cases of Lys and Arg are also an instructive example of how the size of barriers ($U_{max}-U_{min}$) and the hydropathy of the molecule (logP) are not necessarily correlated with each other.

## 4. Discussion

Wrapping up, the systematic study of the toy model and the simulation of the diffusion of amino acids lead us to the following key points:The key determinant for perpendicular diffusion is the overall height of the free-energy barriers. For barrier heights larger than roughly $5k_{B}T$, one finds that $D_{⊥}$ decreases exponentially with the size of the barrier (Figure 3, Figure 5 and Figure 7a).Parallel diffusion is determined by the relative time spent in the lipid phase as compared to water, which provides a direct relation between $D_{\|}$ and the partition coefficient logP (Equation (Equation 14), Figure 6 and Figure 7b);The lower boundary of $D_{\|}$ is equal to $D_{lip}$, obtained for highly-hydrophobic guest molecules. Together with point 1, this indicates that, for large enough barriers, parallel diffusion is dominant.

Experimental assays, such as pulsed-field gradient NMR [26] or macroscopic release setups [17], can access a macroscopic, three-dimensional diffusion coefficient $D_{\mathrm{eff}}$, probing scales equal to or larger than microns. How is $D_{\mathrm{eff}}$ related to the diffusion coefficients considered in the simulations? It is key to observe that, although being ordered at the nanoscale, lipid mesophases are formed by micrometer-sized domains separated by grain boundaries, which usually lack orientational order at larger scales [70]. Hence, assuming neighboring domains to be randomly oriented with respect to each other, there is no net distinction between parallel and perpendicular diffusion at experimentally-relevant scales. The assumed random orientation enables accounting for large-scale diffusion by averaging over the various domains, hence the particles are expected to experience three-dimensional diffusion with an effective diffusion coefficient $D_{\mathrm{eff}}=2D_{\|}/3+D_{⊥}/3$. The weights 2/3 and 1/3 associated with parallel and perpendicular diffusion are chosen so as to account properly for the dimensionality of the corresponding process. For large barriers, one can thus approximate $D_{\mathrm{eff}}=2D_{\|}/3$; by means of Equation (Equation 14), one obtains
(15)$$D_{\mathrm{eff}}=\frac{2}{3}\frac{D_{lip}\left(l\;10^{\log P}+\frac{w}{2}\right)+D_{wat}\left(\frac{a}{2}-l-\frac{w}{2}\right)}{l\;10^{\log P}+\frac{a}{2}-l}\;.$$

Remarkably, Equation (Equation 15) enables estimating the macroscopic diffusion coefficient $D_{\mathrm{eff}}$ from knowledge of local geometrical (a,w,l) and transport ($D_{lip},D_{wat}$) properties, as well as from the overall thermodynamic equilibrium distribution (logP). In Figure 8, we test the accuracy of Equation (Equation 15) by comparing its prediction with the value obtained directly from the simulations, both for the toy model (panel a) and for the amino-acids’ simulations (panel b). For barriers larger than $5\;k_{B}T$, the prediction gives values within 10% of the numerical ones. Nevertheless, it should be kept in mind that the threshold value of the barrier for which the approximation works reasonably is expected to depend on the numerical values chosen for the parameters, particularly for the ratio $D_{lip}/D_{wat}$.

As seen from Figure 8b, in the case of amino acids, the approximation works quite well in virtually all cases. From the literature, typical values of the barriers for various drugs are typically well beyond the threshold value of $5\;k_{B}T$. For instance, molecular dynamics simulations have accessed the potential of mean force for embedding paracetamol in DPPC, obtaining a barrier $≃50\;k_{B}T$ [47]. Similarly, for aspirin and ibuprofen in DPPC, the free-energy barriers have been estimated to be $≃20–30\;k_{B}T$ [71]. These values suggest that, in practical applications, parallel diffusion is often dominant, thus enabling the employment of the approximation given by Equation (Equation 15). Based on this, as a further example, we collected experimental values of logP and $D_{wat}$ for various drugs from the literature, and plugged them into Equation (Equation 15) to estimate the effective diffusion coefficient $D_{\mathrm{eff}}$ characterizing the release from a $L_{\alpha }$ lamellar mesophase with a≃6.7 nm, l≃2.4 m, and w≃1 nm. These values are the geometrical parameters fixed in the toy model, and correspond to experimentally-observed structures for water/DPPC mixtures at 43 °C [46]. The estimations of $D_{\mathrm{eff}}$ are listed in Table 1, and can provide a reference for researchers interested in studying the release of these drugs from lipidic mesophases. Following our treatment, we assume that, for each case, $D_{lip}=0.09D_{wat}$.

Although providing a useful starting point, the results reported in Table 1 are, however, only estimations based on the limited knowledge provided by the experimental partition coefficient. As mentioned above, the ideal strategy is to combine the present approach with atomistic simulations of the system of interest, so as to compute the free-energy profile U(z) and the diffusion profile D(z) to be used as input [69]. As a word of caution, some care has to be taken when dealing with molecules with pK*_a_* values close to the pH of the solution. Indeed, in this case, the possibility of dynamic protonation has to be taken into account. For instance, for the charged amino acids in Figure 7a, we considered for U(z) the value corresponding to the charged or neutral state according to their relative thermodynamic stability, which changes with *z* [63]. The best practice is to run constant-pH atomistic simulations [81], which yield directly the correct free-energy profile [64]. However, we believe that, in practice, this will result in small changes in the predicted transport values, as a molecule with preference for charged states (in pure water) tends to be depleted by the hydropobic center of the lipid bilayer, thus resulting in a high free-energy barrier for bilayer permeation [82]. In turn, this yields negligible values for $D_{⊥}$ (Figure 7b), making parallel diffusion the dominant transport mechanism. While charge neutralization lowers the barrier at the center of the bilayer, it is expected that such barrier is still present and large in magnitude [82]; hence, while promoting perpendicular diffusion, the value of $D_{⊥}$ is still expected to be too small to significantly affect overall diffusion.

To summarize, we have presented a multiscale approach to predict the macroscopic diffusion coefficient by a combination of atomistic simulations, from which the profiles for U(z) and D(z) (Figure 1b,c) can be extracted, and Brownian-dynamics simulations, which enable access to the effective large-scale diffusion emerging from the interplay of the nanoscopic features. Based on a minimalistic toy model and a case study focused on amino acids, we have discerned the impact of the main dynamic and thermodynamic features of the system on molecular transport at macroscopic scales. A further possibility is to use our results the other way round: from experimental knowledge of $D_{\mathrm{eff}}$, $D_{wat}$ and logP, and under the assumption of large barriers and a diffusion profile with the shape considered here (Figure 1c), one can access the value of $D_{lip}$, characterizing diffusion of the inspected molecule within the lipid bilayer. Future work will consider ad hoc studies to obtain the detailed shape of D(z) for selected systems based on established procedures from the literature [69]. Moreover, we will also adapt the present framework to more complex topologies of lipid mesophases of direct relevance for release studies, including, for instance, hexagonal and cubic phases. From a wider perspective, the great potential of multiscale simulations is already being exploited in affine fields, such as the study of biological membranes [83] or the pathways of antibiotic intake from bacteria [84]; it is our hope that the results presented here will spark a similar interest for multiscale simulations in the field of controlled release from lipid mesophases, thus paving the way for the development of an invaluable complement to experimental assays.

## Figures and Tables

**Figure 1 pharmaceutics-15-00573-f001:**
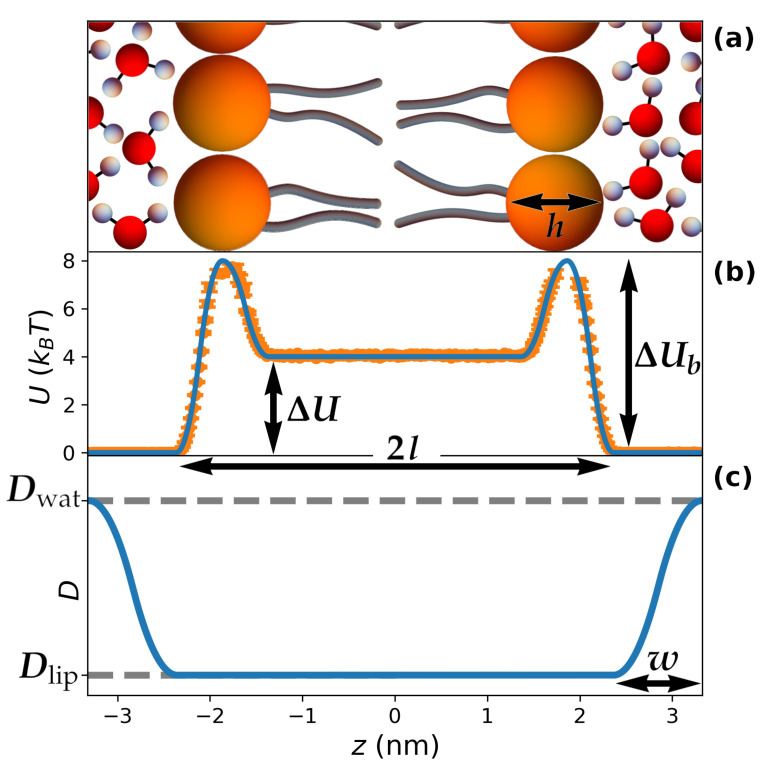
General features of the theoretical model. (**a**) sketch of a repeating unit in a lamellar mesophase and definition of the main parameters; molecules are not at scale; (**b**) representative periodic potential of mean force U(z) corresponding to the system in (**a**). In this case, U(z) corresponds to a hydrophilic molecule (ΔU>0) with low affinity for the lipid heads ($\Delta U_{b}>0$). The orange circles correspond to the free energy extracted from a control simulation to determine the optimal timestep, as described in the Methods; (**c**) representative periodic, position-dependent diffusion coefficient D(z) corresponding to the system in (**a**), smoothly changing from $D_{lip}$ within the lipid phase to $D_{wat}$ in the water phase far from the lipid/water interface.

**Figure 2 pharmaceutics-15-00573-f002:**
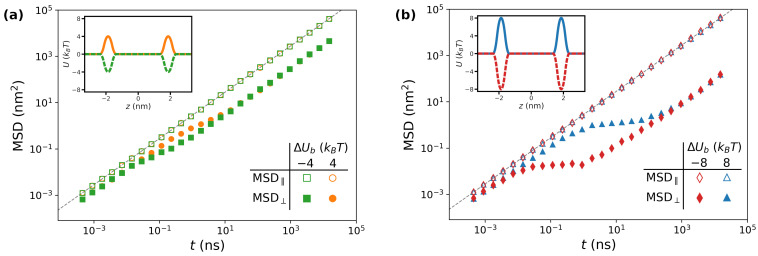
Representative evolution of mean-square displacement as a function of time for $\left|\Delta U_{b}\right|=4$ (**a**) and $\left|\Delta U_{b}\right|=8$ (**b**), while ΔU=0 and $D_{lip}=D_{wat}$. In both panels, filled and empty symbols correspond to $MSD_{⊥}$ and $MSD_{\|}$, respectively, as reported in the legends. The dashed lines correspond to the formula $4D_{wat}t$. In the insets, the potential of mean force for each case is reported.

**Figure 3 pharmaceutics-15-00573-f003:**
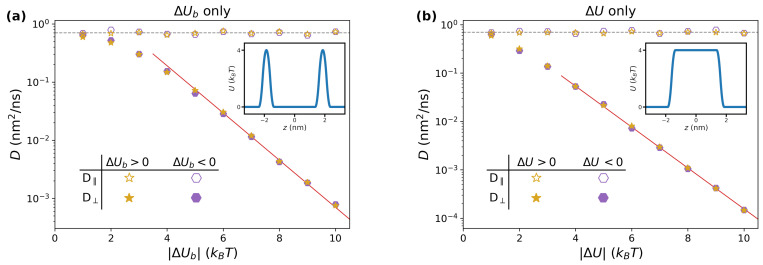
(**a**) Dependence of long-time diffusion coefficients on the size of the barriers $\left|\Delta U_{b}\right|$, for ΔU=0 and $D_{lip}=D_{wat}$. The horizontal dashed line corresponds to $D_{wat}=0.7$ nm^2^/ns. The red continuous line is obtained by fitting the values for $\left|\Delta U_{b}\right|>5k_{B}T$ via an exponential function. In the inset, we report a representative potential of mean force U(z) for this study, corresponding to $\Delta U_{b}=4k_{B}T$; (**b**) same as (**a**), but focusing on varying $\left|\Delta U\right|$ in systems with $\Delta U_{b}=0$ and $D_{lip}=D_{wat}$. In the inset, we report U(z) for $\Delta U=4k_{B}T$.

**Figure 4 pharmaceutics-15-00573-f004:**
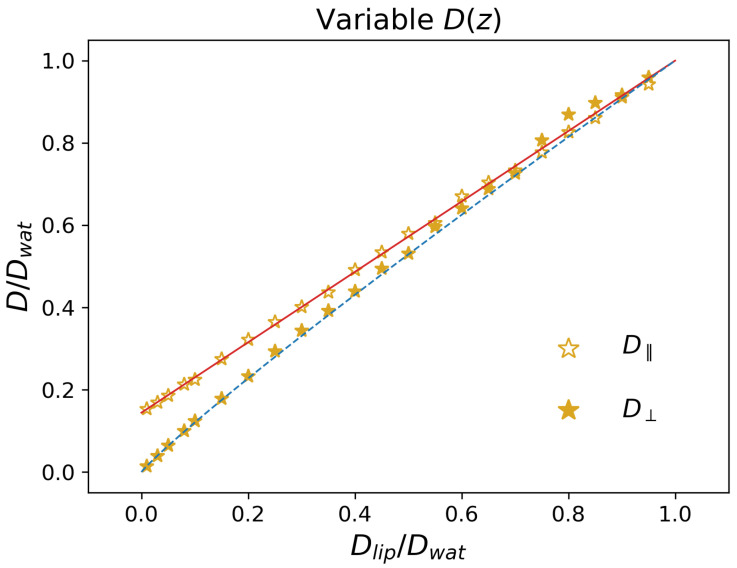
Effective diffusion coefficients in a system with position-dependent D(z) and no external potential. The red continuous line is the prediction from Equation (Equation 12). The blue dashed line corresponds to a fit of the simulation data by a power law.

**Figure 5 pharmaceutics-15-00573-f005:**
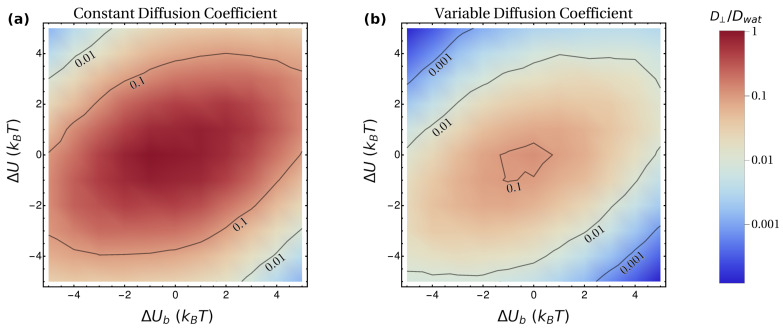
Dependence of the perpendicular long-time diffusion coefficient $D_{⊥}$ on hydropathy (ΔU) and affinity for lipid heads ($\Delta U_{b}$), implemented according to the corresponding potential of mean force U(z) (Figure 1a); (**a**) is obtained by assuming $D_{lip}=D_{wat}$, while (**b**) considers a more realistic diffusion profile with $D_{lip}=0.09D_{wat}$ (Figure 1c). The contours correspond to the indicated values of $D_{⊥}/D_{wat}$.

**Figure 6 pharmaceutics-15-00573-f006:**
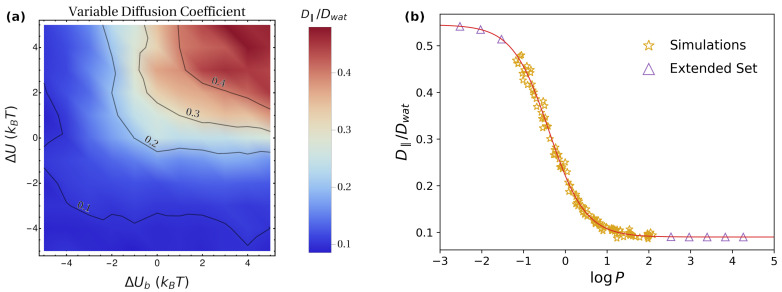
(**a**) Dependence of the parallel long-time diffusion coefficient $D_{\|}$ on hydropathy (ΔU) and affinity for lipid heads ($\Delta U_{b}$), implemented according to the corresponding potential of mean force U(z) (Figure 1a). The diffusion profile corresponds to Figure 1c with $D_{lip}=0.09D_{wat}$; (**b**) dependence of $D_{\|}/D_{wat}$ on logP for the simulation data (golden stars) and for an extended set of systems (purple triangles), for which the lateral diffusion coefficient was computed according to Equation (Equation 13). The extended systems with logP<0 were obtained by considering $\Delta U=\Delta U_{b}=20k_{B}T,200k_{B}T,2000k_{B}T$; the extended systems with logP>0 were obtained by setting $\Delta U=\Delta U_{b}=-10k_{B}T,-9k_{B}T,-8k_{B}T,-7k_{B}T,-6k_{B}T$. The red continuous line corresponds to Equation (Equation 14).

**Figure 7 pharmaceutics-15-00573-f007:**
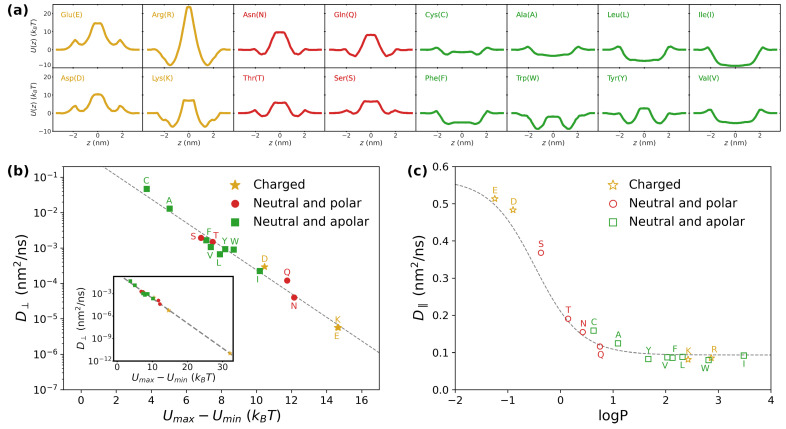
(**a**) Potential of mean force for 16 amino acids as computed in Ref. [63]. The plots were colour-coded according to the physico-chemical properties of the side chain: gold ↔ charged, red ↔ polar and green ↔ apolar; (**b**) perpendicular diffusion coefficient $D_{⊥}$ for charged (filled gold stars), polar (filled red circles) and apolar (filled green squares) residues as a function of the difference $U_{max}-U_{min}$ between maximum and minimum height of the corresponding potential of mean force. In the inset, the full range of $U_{max}-U_{min}$ is considered to include the case of Arg, for which $U_{max}-U_{min}≃32k_{B}T$. The dashed grey line is a best fit via an exponential decay; (**c**) parallel diffusion coefficient $D_{\|}$ for charged (empty gold stars), polar (empty red circles) and apolar (empty green squares) residues as a function of the logarithm of the partition coefficient logP. The dashed grey line corresponds to the theoretical prediction according to Equation (Equation 14).

**Figure 8 pharmaceutics-15-00573-f008:**
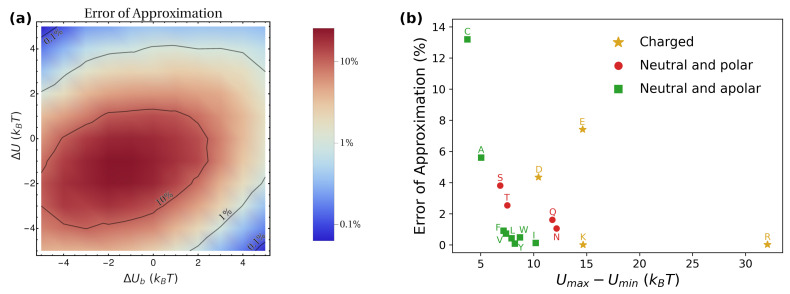
Approximation error on $D_{\mathrm{eff}}$ when employing Equation (Equation 15) for the toy model (**a**) and the amino-acids simulations (**b**). The error is computed as $100\cdot \left|1D_{\mathrm{eff},\mathrm{pred}}/D_{\mathrm{eff},\mathrm{sim}}\right|$, where $D_{\mathrm{eff},\mathrm{pred}}$ is the predicted value according to Equation (Equation 15), and $D_{\mathrm{eff},\mathrm{sim}}$ is computed directly from the simulations.

**Table 1 pharmaceutics-15-00573-t001:** Estimated values of effective diffusion coefficient for release of various drugs from a lamellar mesophase at 43 °C with geometric parameters chosen from the literature. The values of $D_{wat}$ were obtained by renormalizing experimental values obtained at different temperatures via the Stokes–Einstein equation, as discussed in the Methods [57]. Experimental values of $D_{wat}$ and logP were taken from Refs. [42,54,72,73,74,75,76,77,78,79,80].

Name	logP	$D_{\mathrm{wat}}$ (nm^2^/ns)	$D_{\mathrm{eff}}$ (nm^2^/ns)
Cephalexin	−0.67	0.70	0.27
Hydrochlorothiazide	−0.15	1.69	0.43
Levodopa	0.00	0.95	0.21
Piroxicam	0.29	0.85	0.14
Methyldopa	0.39	1.14	0.18
Paracetamol	0.46	1.06	0.15
Antipyrine	1.01	1.04	0.11
Carbamazepine	2.93	1.13	0.10
Ketoprofen	3.31	0.67	0.06
Desipramine	3.94	0.46	0.04
Ibuprofen	3.99	0.77	0.07

## Data Availability

The files containing the data used to generate the plots are available as Appendix A.

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
