# Peer review of "Interplay of Hydropathy and Heterogeneous Diffusion in the Molecular Transport within Lamellar Lipid Mesophases"

_pharmaceutics, 2023, doi:10.3390/pharmaceutics15020573_

Round 1
Reviewer 1 Report
This is well written manuscript.
However, It would have been nice to see the variation of diffusvity with its hydropathy index of the amini acids studied.
In addition glycine being the shrotest amino acid, be tested as the reference case.
Reviewer 2 Report
The paper deals with the diffusion through the biological membrane. The paper seems to me quite interesting, Especially the summary data of all amino acids. However, the constitution of model is unclear.
I am very question why the behavior of glutamine and Asparagine differ from Lysine and Arginine? It is very difficult to answer this and more other questions.with regard of the model. It is because I cannot imagine, how the model works and how the model structure looks like. That is why I require major revisions (i.e. detailed descxription of the simulation).
11. Which level of coarsening did you use? Did you use Fully atomistic/ United atom/ Coarse graining/ Dissipative particle dynamics or other approximation? Please provide the information together with snapshot of simulation.
22. How the amino acids are modeled? Are they fully atomistic or some simplification? Or one molecule=one bead?
33. How you provide the cohesion of the lipid bilayer? When one simulates such orgainzed structure by dynamic simulation, the structure is usually disordered during simulation. Was there used function freeze selected atoms or did you use some contraints?
44. What type of boundary conditions was applied? Were applied the periodic in xy direction?
After I will get the detailed information, I will be able to evaluate the discussion of model outputs.
Reviewer 3 Report
The manuscript “Interplay of hydropathy and heterogeneous diffusion in the molecular transport within lamellar lipid mesophases” by Antonio M. Bosch et al. has established a phenomenological model to evaluate how hydropathy and diffusivity may influence the permeation through lipid bilayers. Overall, I find this paper an excellent theoretical study of permeation with a well-structured and clear presentation. I believe this is a high-quality paper that merits publication inPharmaceutics. I only have some minor questions for the authors to address.
1. The authors set values for parameters for Equation 1 based on the structural features of DPPC lipids. Then what about other lipid molecules? It should be better if the authors give some guidelines for the parameterization. E.g., I guess “l” and “l–h” respectively correspond to the bilayer thickness (DB/2) and hydrophobic thickness (DC) of a lipid bilayer, both of which can be determined by small angle neutron or X-ray scattering. If so, the authors should explicitly indicate that those experimental measures are what others should search for if they want to apply the model to other lipids.
2. The authors intraplate the space-dependent diffusivity of a permeant, D(z), from Dwat and Dlip. How should Equation 2 be updated for permeants with a D(z) that falls beyond this range?
3. In the applications of amino acids, the authors employed the U(z)’s from existing free energy calculations. It is well known that the charge states (e.g., Asp–1 vs. Asp0) may greatly impact the permeation free energy profile (or PMF, see J. L. MacCallum, et al. Biophys. J. 2008, 94, 3393–3404; A. C. Johansson, et al. J. Phys. Chem. B 2009, 113, 245–253; D. Bonhenry, et al. J. Chem. Theory Comput. 2013, 9, 5675–5684; D. Bonhenry, et al. Phys. Chem. Chem. Phys. 2018, 20, 9101–9107). But the authors only chose the PMF derived using the neutral form. Please be advised that all these calculations (and thus the authors’ modeling) assumed no protonation during the permeation. This assumption is crude for ionizable permeants, e.g., Asp, Glu, and Lys. Because a permeant is likely to change its charge state during the permeation (V. H. Teixeira, et al. J. Chem. Theory Comput. 2016, 12, 930–934; B. K. Radak, et al. J. Chem. Theory Comput. 2017, 13, 5933–5944). The resulting PMF with this dynamic protonation can differ significantly from those assuming a constant charge state (Z. Yue, et al. J. Am. Chem. Soc. 2019, 141, 13421–13433). Permeation of ionizable drugs is quite similar (J. Ulander, et al. Biophys. J. 2003, 85, 3475–3484; Q. Zhu, et al. Sci. Rep. 2017, 7, 17749; Z. Yue, et al. J. Am. Chem. Soc. 2019, 141, 13421–13433). I think this can be a limitation of the current works. Thus, how would considering this dynamic protonation impact the results presented in Figures 7 and 8?
Round 2
Reviewer 2 Report
The authors significantly improved the manuscript and answered the questions and comments. I recommend this paper for publication.